# Living with *Legionella* and Other Waterborne Pathogens

**DOI:** 10.3390/microorganisms8122026

**Published:** 2020-12-18

**Authors:** Joseph O. Falkinham

**Affiliations:** Department of Biological Sciences, Virginia Tech, Blacksburg, VA 24061, USA; jofiii@vt.edu

**Keywords:** premise plumbing, disinfectant-resistant, oligotrophic, stagnant, biofilm formation

## Abstract

*Legionella* spp. and other opportunistic premise plumbing pathogens (OPPPs), including *Pseudomonas aeruginosa*, *Mycobacterium avium*, *Stenotrophomonas maltophilia*, and *Acinetobacter baumannii*, are normal inhabitants of natural waters, drinking water distribution systems and premise plumbing. Thus, humans are regularly exposed to these pathogens. Unfortunately, *Legionella* spp. and the other OPPPs share a number of features that allow them to grow and persist in premise plumbing. They form biofilms and are also relatively disinfectant-resistant, able to grow at low organic matter concentrations, and able to grow under stagnant conditions. Infections have been traced to exposure to premise plumbing or aerosols generated in showers. A number of measures can lead to reduction in OPPP numbers in premise plumbing, including elevation of water heater temperatures.

## 1. Legionella and Other Opportunistic Premise Plumbing Pathogens (OPPPs)

*Legionella pneumophila* is a member of a group of waterborne opportunistic pathogens that surround humans. Others in the group include the bacteria: *Pseudomonas aeruginosa, Mycobacterium avium* complex (MAC), *Mycobacterium abscessus*, *Stenotrophomonas maltophilia*, and *Acinetobacter baumannii*, and the amoebae, namely, *Acanthamoeba* spp. Collectively, they are called opportunistic premise plumbing pathogens (OPPPs). Humans are continually exposed to these opportunistic pathogens as they colonize and persist in drinking water in homes, apartments, schools, hospitals, and offices.

The infections caused by the OPPP can be in the lungs, skin, or bloodstream depending upon the point of entry of the infecting microorganism. They are called opportunistic pathogens as they infect that proportion of the population with risk factors. Those risk factors include older age, injury, surgery, immunodeficiency due to infection (i.e., HIV), cancer, or chemotherapy, and prior lung damage due to infection, smoking, or exposure to dusty occupations. The infections can be acquired in hospitals (nosocomial infections) or in the community at home, at work, at play. For example, *P. aeruginosa* wound and urinary tract infections occur in hospitals and present treatment problems because of *P. aeruginosa’s* intrinsic resistance to antibiotics and disinfectants [1] and *L. pneumophilia* lung infections have been linked to hotel stays [2]. *Acinetobacter baumannii* was found responsible for infections in injured U.S. troops in the Middle East [3].

Infections caused by *L. pneumophila*, a life-threatening pneumonia, and Legionnaires’ disease must be reported to health officials, so we know that there were at least 10,000 cases in the United States in 2018 [4]. As for the other OPPPs, infections are not required to be reported, so the prevalence and incidence of disease are estimates. Particularly for infections caused by *M. avium*, the estimated numbers are quite high; namely, 180,000 in the United States [5]. The estimated cost of Legionnaires’ disease in hospitalized patients in the United States in 2012 was over USD 433 million and the cost of nontuberculous mycobacteria (NTM) disease of which a majority was due to *M. avium* was over USD 425 million [6]. In 2017, the CDC reported 32,600 estimated cases with 2700 deaths due to *P. aeruginosa* infections in hospitals with an estimated cost of USD 767 million [7].

## 2. Common Characteristics of *Legionella* and OPPPs

### 2.1. Colonists, Not Contaminants

The first common, shared characteristic of *Legionella* and the other opportunistic premise plumbing pathogens (OPPPs) is that they are all colonists of drinking water, not contaminants. Contaminants, such as the fecal bacterium *Escherichia coli*, do not grow or survive in drinking water, they die the further away they get from the site where the water was polluted. In contrast, *Legionella* and the other OPPPs survive, grow, and persist in natural and drinking water; that is their home. For example, the numbers of *M. avium* increase in the distribution system between the water treatment plant and homes, offices, and hospitals [8]. Further, *P. aeruginosa* was shown to grow in hospital distilled water [9].

Natural soils and surface waters are the ultimate sources of *Legionella* and the other OPPPs. They are adapted to growth and survival in flowing water systems and are particularly adapted to survival, persistence, and growth in the “built environment”, particularly in hospital and household plumbing.

### 2.2. Disinfectant-Resistance

The second common character of the OPPPs is their disinfectant-resistance. In the United States, water from surface rivers and lakes is transported by gravitation or pumped to treatment plants where total microbial and pathogen counts are reduced by a series of steps, including precipitation and disinfection. The most common disinfectant used in the United States is chlorine which is an effective killer of microorganisms—except the OPPPs. *M. avium*, *P. aeruginosa*, *S. maltophilia*, and *A. baumannii* are significantly more resistant to chlorine than is Escherichia coli, the bacterial standard used to gauge disinfection efficacy. Strains of P. aeruginosa, particularly the mucoid variants, are chlorine-resistant [10] as are multidrug-resistant *A. baumannii* isolates [11]. *M. avium* is approximately 1000-fold more tolerant to chlorine than is *E. coli* [12]. At a concentration of 1 mg chlorine per mL, which is used widely in the United States for disinfection, 99.9% of *E. coli* cells are killed in 5 min. In contrast, to kill 99.9% of *M. avium* cells, the exposure duration would have to be above 80 h. Although *M. avium* is an extreme example of disinfection-resistant waterborne bacteria, the other OPPPs are also resistant. The consequence of disinfectant-resistance amongst the OPPPs is that they are not killed by water treatment processes and enter the water distribution systems where they grow and are transported to homes, apartments, hospitals, schools, and elder care facilities. The OPPPs have no competition for nutrients from the chlorine-sensitive fecal contaminants in water, so all, even though limited, organics are available to *L. pneumophila*, *M. avium*, *P. aeruginosa*, *A. baumannii*, and the other OPPPs.

In addition to disinfectant-resistance, there is an interesting coincidence of disinfectant- and antibiotic-resistance amongst the OPPPs. This has been particularly investigated in *P. aeruginosa* where isolates from patients and hospital water and wastewaters are resistant to both antibiotics and disinfectants [13]. One reason for the simultaneous resistance is the high number of drug-efflux pumps in *P. aeruginosa* [13]. Rather than efflux pumps, resistance of *M. avium* to antibiotics and disinfectants is due to the presence of the thick, lipid-rich outer membrane of *Mycobacterium* spp. that provides an impermeable layer to hydrophilic compounds such as antibiotics and disinfectants [14].

### 2.3. Growth in Natural Water, Distribution Systems, and Premise Plumbing

As noted above, *L. pneumophilia* and the other OPPPs are colonists of human-engineered water systems. Their source is natural waters and soils where they have been subject to selection for growth under low nutrient conditions. Thus, these opportunistic pathogens are quite able to survive and proliferate in plumbing in hospitals and homes. They grow and survive under a wide range of conditions, including high temperature (e.g., water heaters) and low oxygen (e.g., stagnation). *P. aeruginosa* was shown to persist in premise plumbing of buildings [15,16], *S. maltophilia* was recovered from hospital tap water [17], and Acinetobacter spp. has been isolated from rural water supplies [18]. *M. avium* was shown to grow under low-nutrient, oligotrophic conditions in a pilot water distribution system [19]. Growth in water distribution systems means that the numbers of *M. avium* increase as the water is transported through the distribution system pipes to homes and hospitals [8]. Unlike *E. coli* and other enteric pathogens, OPPP numbers increase from the source due to the inability of those fecal pathogens to persist in the low nutrient conditions of drinking water. Passage through the water heater provides an opportunity for growth of *L. pneumophila* and *M. avium* and stagnation leads to increases in OPPP numbers as many have means to grow under microaerobic conditions as is the case for *M. avium* [20], or even anaerobiosis in the case of *P. aeruginosa* that can use nitrate (NO_3_^−^) as a terminal electron-acceptor in the absence of oxygen [21].

### 2.4. Surface Adherence and Biofilm Formation

Another shared characteristic of *L. pneumophila* and the other OPPPs is their propensity to attach to surfaces where they grow to form biofilms—layers of cells imbedded in an extracellular matrix. All OPPPs readily adhere to surfaces and form biofilms including: *L. pneumophila* and *P. aeruginosa* [22], *B. cepacia* [23], *S. maltophilia* [24], and *A. baumannii* [25]. Surface attachment and biofilm formation are positive survival responses to flowing systems, such as rivers, drinking water distribution systems, and premise plumbing. Adherence prevents cells from being washed out. Further, the growth of attached cells is accompanied with the production of an extracellular matrix in which the OPPP cells are embedded. Depending on the OPPP species, the extracellular matrix can be rich in polysaccharides (e.g., *P. aeruginosa*) or lipid (e.g., *M. avium*). Biofilms are formed in both plumbing and in infected patients. The extracellular matrix serves as a barrier to exposure of cells to antimicrobial agents; namely, antibiotics in infected individuals and disinfectants in drinking water systems [26]. Antibiotics [27] and disinfectants, such as chlorine [28], are unable to penetrate the matrix, and cell growth in biofilms modifies their susceptibility to antimicrobial agents [12] so that cells can survive and grow in biofilms. The extracellular matrix also reduces the penetration of oxygen into biofilms [29], so the majority of biofilm cells are growing under microaerobic or anaerobic conditions as the spatial distribution of *P. aeruginosa* in biofilms is governed by oxygen concentrations [30].

The star performers of biofilm formation are the members of the *Mycobacterium* genus, namely, *M. avium*, *M. chimaera*, and *M. abscessus*—the major human pathogens. *Mycobacterium* spp. cells are surrounded by a long chain (C_60_-C_80_), lipid-rich outer membrane [14] that makes the cells hydrophobic. For example, water beads on the surface of mycobacterial cells collected on a filter. The consequence is that cells of all three species rapidly adhere to surfaces to avoid the high polar charges of water. Within 1 h of exposure to a cell suspension of 1000 *M. avium* or *M. abscessus* cells/mL, greater than 1000 cells/cm^2^ can be detected on stainless steel, PVC, glass, and galvanized surfaces [31]. Further, surface adherence is quite rapid because inoculation of the 15 L water reservoir of a heater–cooler with 10 billion *M. chimaera* cells resulted in an apparent loss of 99.9% of the cells within 5 min due to their adherence on the pipes, tubes, and walls of the heater–cooler [21]. Similarly, *P. aeruginosa* is also an avid biofilm former [32].

### 2.5. Amoebae-Resisting Microorganisms

*L. pneumophila*, *P. aeruginosa*, *M. avium* complex, and the other OPPPs are readily phagocytozed by amoeba, including Acanthamoeba, Vermamoeba, and Hartmanella [31,32] and protozoa including Tetrahymena [33]. One reason for the presence of OPPPs in amoebae and protozoa is that they share the same drinking water environment [33,34]. OPPPs within amoeba or protozoa can be detected and enumerated by fluorescent microscopy [35] or by colony formation or PCR in mild detergent-lysed amoeba or protozoa. Free-living amoeba and protozoa are constantly grazing on microorganisms in biofilms and utilizing them as nutrient. In the case of the OPPPs, phagocytosis does not result in the death and consumption of the OPPPs as nutrient, but their survival and growth; thus, the names amoeba-resisting microorganisms (ARMs) or amoeba-resisting bacteria (ARBs). Resuscitation of viable but unculturable (VBNC) *L. pneumophila* was demonstrated in Acanthamoeba castellanii [36]. As was pointed out above using the examples of growth at low nutrient concentration, disinfectant-resistance, growth at low oxygen levels, and growth in biofilms, resistance to digestion in amoeba or protozoa is a mode of survival for OPPPs in soils, natural waters, and human-engineered water systems. Further, it has been hypothesized that the survival and growth of *L. pneumophila* and other OPPPs in human phagocytic cells is a direct consequence of selection for survival and growth of ARMs in amoeba and protozoa [33,34].

Amoeba and *L. pneumophila* share the same environments and coincident isolation of amoeba and *L. pneumophila* are frequently reported. Whether that coincident recovery of both amoeba and *L. pneumophila* is an indicator of obligate dependence of *L. pneumophila* for survival, persistence, and growth in drinking water or is simply a result of microorganisms sharing the same habitats remains to be proven. This sharing of the same environments, namely, human-engineered water systems, is found with *M. avium* and the other OPPPs. However, *L. pneumophila*, *M. avium*, *P. aeruginosa*, *A. baumannii*, and *S. maltophilia* are found in the absence of amoeba and protozoa. It would be valuable to identify biomarkers of OPPPs that are signs of prior growth in amoeba or protozoa. For example, would there be changes in mycobacterial outer membrane lipids or proteins originating from the amoeba or protozoa? Such markers would permit further examination of the relationship between free-living amoeba and protozoa and *L. pneumophila* and the other opportunistic premise plumbing pathogens.

It is possible to mix drinking water samples with *Acanthamoeba polyphaga* to increase the species variety of *Mycobacterium* spp. isolates. Following inoculation of a water sample with and without starved cells of *A. polyphaga* and incubation for 4 days, isolation of triple-washed amoeba cells lead to recovery of a greater variety of *Mycobacterium* species isolates compared to the un-inoculated control (Falkinham, unpublished).

The association between OPPPs and amoeba and protozoa may be mutually beneficial. Tetrahymena spp. fails to grow from low-density inocula unless the medium was supplemented with a variety of compounds, including fatty acids [37]. However, a derivative of *T. pyriformis* carrying *M. avium* was able to grow from a low-density inoculum (Falkinham, unpublished). Possibly, the intracellular *M. avium* cells supplied fatty acids to *T. pyriformis* permitting growth from low cell densities.

### 2.6. Interactions amongst Legionella and OPPPs

In addition to the interaction between free-living amoeba and protozoa and *L. pneumophila*, *M. avium*, and other OPPPs, there are several other interactions that have been observed between *P. aeruginosa*, *Mycobacterium* spp., and *L. pneumophila* that deserve further exploration [22,38].

An example of interaction between waterborne opportunistic pathogens is the exclusion of *Mycobacterium* spp. by the salmon-pink pigmented Methylobacterium spp. [39,40]. Specifically, the two bacterial groups are not found in the same biofilm samples. It appears the basis for the mutual exclusion is that occupation of a surface by one, prevents the adherence and biofilm formation by the other [41]. Both are common shower bacteria [39,42] and therefore would be likely to coinhabit the same premise plumbing.

Pseudomonas aeruginosa produces a wide range of secondary metabolites that are antimicrobial [43]. For example, the fluorescent blue-green pigment pyocyanin is strongly antimicrobial. It is possible that pyocyanin in drinking water distribution systems and premise plumbing is capable of reducing numbers of other OPPPs, for example, *L. pneumophila* [44], or stimulating changes in the premise plumbing microbiome to reduce OPPP numbers indirectly. The presence of other OPPPs, for example, Flavobacterium spp. and Pseudomonas fluorescens, has been shown to increase persistence of *L. pneumophila* in biofilms [44].

## 3. Premise Plumbing—An Ideal Habitat for *L. pneumophila* and Other OPPPs

*L. pneumophila*, *M. avium*, *P. aeruginosa*, and other OPPPs love premise plumbing; it is an ideal habitat for their survival, growth, and persistence. First, premise, as it is farthest from a treatment plant, has a low residual disinfectant concentration. Enough perhaps to kill and inhibit growth of fecal coliforms, but not enough disinfect to kill or inhibit the growth of *L. pneumophila*, *M. avium*, *P. aeruginosa*, and other OPPPs. Second, pipes in plumbing offer a high surface to volume ratio that is ideal for surface adherence and biofilm formation. A common feature of OPPPs is their predilection for surface attachment and growth in biofilms. The rather slow growing *L. pneumophila*, *M. avium*, *P. aeruginosa*, and other OPPPs attach to surfaces and therefore do not get washed out of the plumbing. Further, biofilm protects microbial cells from whatever disinfectant is present. Third, sediment collects in the water heater and provides nutrient for microorganisms. Fourth, water in premise plumbing is regularly heated (i.e., 50 °C water heater) and distributed throughout the structure (home, apartment complex, hotel, business, hospital, or healthcare facility) where it cools to an optimal temperature for OPPP growth (e.g., 25–37 °C). Recirculating hot water systems in apartment and condominium buildings and hospitals provide optimal temperatures for growth of *L. pneumophila* and OPPPs. For example, highest numbers of *M. avium* (e.g., 100,000 colony-forming units/mL) have been recovered from water samples collected from high-rise apartments and condominium buildings in New York City [45]. Fifth, the water age in any building is not always the same over time. For example, in hospitals where wards or wings are not always occupied, water ages within that portion of the distribution system. The same occurs in our homes when the children grow up and leave home and their bathrooms and showers are not used for months during school terms. As water ages, disinfectant concentrations fall and OPPPs multiply, growing on available nutrient. Water age has become a critical problem in offices and schools during the COVID-19 pandemic. Buildings and schools have been unoccupied for months to prevent high density gathering of individuals. However, while buildings are closed, the OPPPs continue to grow. If water is immediately used upon re-opening, individuals will be exposed to high numbers of OPPPs. As a consequence, plans need to be developed for opening closed buildings. At the least, thorough flushing of the entire building plumbing is required. That is not only important to consider for building owners and operators, but also for individuals who have second homes. While individuals are at a primary residence, the water is aging in their second home. Therefore, all second homeowners must have plans to flush the entire water system of a second home; keeping in mind that upon leaving the second home, the primary home has older water and requires flushing. Sixth, water flow in premise plumbing, whether a home, apartment, condominium, hospital, or healthcare center is intermittent. In homes, hot water flows through the plumbing in mornings with bathroom use and showers and in evenings with kitchen cleanup. Hot water flow is coincident with human activity. When water flow halts, after we go to school or work from home, or after morning cleaning, water is not moving and stagnates. Microorganisms grow and consume oxygen as they metabolize nutrients. However, as pointed out above, a common feature of *L. pneumophila* and other OPPPs is that they all grow at low oxygen levels (e.g., microaerobic). As the water in mornings has been heated, conditions are ideal for growth in household, hospital, and healthcare plumbing. Remember, while we humans are away, the microbes grow in plumbing and reach high numbers.

## 4. Pathways of Transmission of Legionella and OPPPs

Pathways of transmission of OPPPs include: aerosolization, drinking water, and coming in contact with water. Humans are exposed to OPPPs through all three pathways at home, at work, or in hospitals. Aerosolization refers to the generation of water droplets from water. Most commonly, those aerosols are inhaled leading to lung infections. As the water could carry *L. pneumophila*, *M. avium* [46], *P. aeruginosa* [47,48], *A. baumannii* [49,50], or other OPPPs, any source of aerosols could be an OPPP source. In homes, apartments, and condominiums, aerosols are generated by showers [46,51], humidifiers [52], aerators on taps and faucets [47], from drains and from hot tubs and spas [53], fish tanks (aquaria), and watering indoor plants.

Contact with water colonized with OPPPs can lead to infection. Solutions of skin surface disinfectants can become colonized by OPPPs, particularly *P. aeruginosa* [54]. Due to its broad resistance to antimicrobials and presence in water, even distilled water, *P. aeruginosa* can persist in sterilizing solutions and be introduced to skin or wounds directly [54]. In a similar fashion, an *S. maltophilia* hospital infection outbreak was traced to a *S. maltophilia*-colonized disinfectant solution [55]. Hospital tap water has been identified as the source of *S. maltophilia* infection in a hospital [17].

The home environment can have unexpected sources of OPPPs. Kitchen sponges have been shown to harbor “massive” colonization of Acinetobacter spp. [56]. Chilled water from a refrigerator was shown to be the source of M. abscessus infection in an elderly couple (Falkinham, unpublished).

Sources of OPPPs abound in hospitals and healthcare centers. In addition to patient room showers and sinks, patients are exposed to aerosols generated by therapy baths, patient manipulations, and medical equipment. Medical equipment that has been shown to be aerosol sources of OPPPs include bronchoscopes and heater–coolers. Heater–coolers are water heating and cooling devices used in operating rooms to cool patients and warm blood during periods of time when the heart may be stopped and oxygen–carbon dioxide exchange is performed outside the body. In 2015, one manufacturer’s heater–cooler was shown to be the source of *Mycobacterium chimaera* aerosols that infected patients undergoing cardiac surgery [57]. Although the prevalence of the infections was low throughout the world, the mortality was high at 50%. Eventually, it was shown that the device aerosolized *M. chimaera* cells that had colonized the water cycle of the instrument. The *M. chimaera* strain was introduced during final testing of the instrument before transmission.

## 5. Monitoring for *Legionella* and OPPPs

In light of the burden for medicine to do no harm and the enormous cost of *L. pneumophila*, *M. avium* complex, and other OPPP infections [6], it is of importance to consider measuring numbers of OPPPs in drinking water, especially in structures where a significant proportion of individuals have risk factors for OPPP infection, namely, hospitals. However, as *L. pneumophila, M. avium P. aeruginosa* and other OPPP infections occur in both community and healthcare settings, how is monitoring to be implemented?

Although *L. pneumophila* infections require notification to public health officials, only New York City, New York, and the Veterans Administration require monitoring of *L. pneumophila*. No monitoring is required for the other OPPPs. Upon discovery of outbreaks, samples are collected to identify OPPP sources and count numbers, but with the exception of *L. pneumophila*, the numbers are not of particular value. Source tracking is of value, but in the absence of well-established dose–response values for *M. avium, P. aeruginosa, A. baumannii*, and *S. maltophilia* infections, there is no guidance for action levels. Further, enumeration of *E. coli*, fecal coliform, and heterotrophic plate count bacteria numbers do not serve as surrogates of OPPPs [58,59]. Perhaps monitoring OPPPs as a general class could be accomplished by following *P. aeruginosa* as a surrogate indicator for OPPP presence [60].

Two other problems emerge when considering monitoring for OPPPs. First, what samples should be collected (e.g., water or biofilm; hot or cold; first draw or delayed draw)? Second, as other microorganisms are present in drinking water, should samples be disinfected and what disinfectants should be employed? Decontamination is widely employed for *Mycobacterium* spp. cultivation due to their ridiculously slow growth and the loss of samples due to overgrowth by other microorganisms. However, disinfectants do kill *Mycobacterium* spp., so their use reduces the sensitivity of detection. Further, even in *Mycobacterium* spp., different species have significantly different susceptibilities to disinfection [61].

## 6. Remediation of *Legionella* and OPPPs in Premise Plumbing

There are two approaches to reducing the impact of *L. pneumophila, M. avium complex, P. aeruginosa, A. baumannii*, and *S. maltophilia* infections on public health, whereby prior planning and remediation after outbreaks are required. Both can be effective. First it is important to point out that the OPPPs share many characteristics in common, such that their numbers reflect reactions to similar conditions. Those shared characteristics can identify measures to design away from constructing OPPP-friendly buildings or perform remediation to reduce OPPP numbers in existing structures. Although it has not been proven for all the OPPPs, it is hypothesized that reductions in the numbers of *L. pneumophila, M. avium, P. aeruginosa, A baumannii*, and *S. maltophilia* will consequently reduce the number of OPPP infections. It is with that unproven hypothesis that the following recommendations are presented.

Homes, apartment buildings, condominiums, hospitals, and health care facilities can be designed to avoid creating ideal growth centers for OPPPs. The current interest in “Green Buildings” has been challenged by listing the green building characteristics that will actually promote OPPP growth and persistence [62]. For example, lower hot water temperatures would likely promote OPPP numbers as will reduced water use resulting in increased water age [63]. The current inclusion of bathrooms in every hospital patient room, while convenient, exposes patients to aerosols from showers, sinks, and drains. Recirculating hot water may save the expense of installing individual water heaters in each patient room, but results in high circulating numbers of OPPPs. Pipe materials can be chosen to reduce OPPP adherence and biofilm formation, thereby reducing the microbial load in the distribution system.

There are a variety of measures that might reduce OPPP numbers based on logic. One remediation measure has been tried and proven to reduce numbers of *M. avium*. It was noted that homes with water heater temperatures equal to or lower than 125 °F (50 °C) had *Mycobacterium* spp., but in homes with water heater temperatures at 130 °F (55 °C) or higher seldom had *Mycobacterium* spp. [63]. Based on that background data, 10 collaborating *M. avium* patients in Wynnewood, Montgomery County, PA, allowed us to raise their water heater temperatures to 130 °F (55 °C). Numbers of *M. avium* in their water samples were measured and, by 12 weeks, 10 out of 10 households no longer had *M. avium* [64]. Thus, one way to reduce the numbers of *M. avium* and quite possibly *L. pneumophilia* and other OPPPs is to raise water heater temperatures to 130 °F (55 °C) or higher.

As showerheads have been shown to be rich in *Mycobacterium* spp. [39,46] and with the presence of *Mycobacteriu**m* spp. in showerheads being the strongest predictor of *Mycobacterium* spp. infection in residents [51], it has been recommended to unscrew, clean and disinfect showerheads monthly by submerging in 5–6% sodium hypochlorite. Further, to reduce the formation of OPPP-rich shower aerosols, install a showerhead with large holes (e.g., >2 mm) to produce large drops which do not enter the alveoli of the lungs, do not form a mist, and whose drops fall rapidly. The water in unused or seldom used wings of a house of rooms should be flushed regularly to reduce the water age and stagnation to keep OPPP numbers low. Microbiological filters with pore sizes of less than 0.22 micrometers in diameter can be installed on showers and taps to produce bacterial-free water.

Granular activated charcoal (carbon) (GAC) filters, which are effective at removal of chlorine, metals and organics, do not prevent the passage of OPPPs, as the pores are too large. In fact, a study at EPA showed that *M. avium* and other *Mycobacterium* spp. grew in GAC filters, such that the GAC filter became the source of *M. avium* in the water [65]. One can use a GAC filter, but its water must be filtered through a 0.22-micron meter pore size filter. Finally, aerosol generating devices should be removed—namely, hot tubs and spas, therapy baths, indoor pools, and “ultrasonic” humidifiers. All can generate high density OPPP aerosols.

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
