# Peer review of "Living with Legionella and Other Waterborne Pathogens"

_microorganisms, 2020, doi:10.3390/microorganisms8122026_

Round 1
Reviewer 1 Report
This a good review on Legionella and other waterborne related pathogens.
As minor revisions: The section on ARB could be further detailed to provide more information to readers. i.e. most common amoeba carrying ARB, protocols used so far etc. Nevertheless it is a good ms and merits publication in the journal.
Author Response
I thank this Reviewer for their kind remarks and suggestion to expand the information in the section on amoeba-resisting bacteria (ARBs).
The information added identifies methods to enumerate and isolate ARB from amoeba and protozoa and the ability of amoebae to resuscitation viable, but unculturable (VBNC) L. pneumophila (with reference).
Reviewer 2 Report
Overview:
The purpose of this manuscript was to discuss Legionella and opportunistic premise plumbing pathogens by comparing their living conditions and interactions and ways to monitor and remediate. The manuscript provides a good review of the topics in general; however, some ideas have room to be further supported and more detail provided about Legionella specifically.
Major Comments:
- The manuscript is presented as focusing on Legionella and other OPPPs, however in a number of sections the only relevant background resources are about Mycobacterium species when similar information about Legionella and/or Pseudomonas exists in the literature. These sections should be expanded upon to include Legionella (Sections as noted: 2.2., 3.2., 3.3., 5.0., 6.0.)
- Lines 125-128: note “Do not sample water. Rather, sample biofilms.” While OPPPs may flourish more in biofilms, it is unrealistic and to recommend sampling only biofilms and irresponsible to advocate against sampling water. These biofilms may be growing inside fixtures and at any points along premise plumbing piping. However, sampling biofilms in the field beyond the outlets is difficult and biofilm sampling does not represent the actual exposure risk of people who would be exposed to that water as an OPPP source. Environmental monitoring for OPPPs in water is an attainable and simple method to determine exposure risk to water in a building water system.
- Lines 142-144: Referring to “there remains the question of whether pneumophila is able to grow freely in water or is dependent upon amoeba.” Amoeba aid in the persistence of Legionella, but not all Legionella are completely dependent upon amoeba for survival. Citations needed to support original argument. In laboratory settings, Legionella are able to grow in broth cultures and even in sterile DI water without the addition of amoeba so it is unlikely this is a remaining question.
- Lines 151, 157-171, 181, 184, 246: Writing switches to first person point of view. Suggest keeping point of view consistent throughout; this approach detracts from the manuscript as a review and not research article
- Lines 176-179: Please clarify exclusion. “exclusion of Mycobacterium by the salmon-pink pigmented Methylobacterium spp.” contradicts “basis for the mutual exclusion.” Additionally, Legionella and Pseudomonas have a demonstrated competitive relationship in cultures and biofilms which should be addressed here.
- Section 5.0. discusses three pathways of transmission but focuses only on aerosolization. There are references noting wound infections with most of these organisms, but this is not covered and the specifics in this section do not reference Legionella. Legionella is a source of hospital-acquired infections transferred by medical devices, water/aerosols, skin/wound infections, catheter-associated infections, etc. These should be addressed here.
- Section 6.0:
- “In light of the enormous cost…” This sentence stresses the cost of infections being the main reasoning behind monitoring OPPPs in hospitals, whereas the ethical implications of not working to prevent infections in at-risk individuals should be emphasized. Consider citing number or cases and/or deaths and expanding upon this.
- There are monitoring requirements for Legionella in New York State and New York City as well as all VHA healthcare facilities. This should be noted.
- The author notes that there is not an established dose-response value for other OPPPs, other than Legionella but the dose-response for Legionella has not been well established either. Please clarify or provide additional information.
- Studies have been published looking at the relationship between Legionella and Pseudomonas and HPC not just fecal bacteria, consider addition.
Minor Comments:
- Section and subsection numbering:
- Line 20: “1.0.”
- Follow journal guidance for numbering of sections and subsections: Lines 47, 60, 86, 101, 129, 172, 193, and 234 include a decimal at the end, while lines 20, 46, 261, and 284 do not.
- Between lines 60 and 86, it jumps from section 2.2 to 3.2 without a main title for 3.0 or a subsection 3.1. Please clarify or consider renumbering. All subsequent numbered sections and subsections should be updated accordingly.
- Change pneumonia to L. pneumophila in lines 34, 87, and 313
- Line 10: “are normal inhabitants of natural waters, and drinking water distribution systems, and premise plumbing.”
- Line 12: “features that allows them”
- Line 33: “infections occur in hospitals and presents treatment”
- Line 34: clarify “its” refers to “ aeruginosa” and not “infections”
- Line 37: “… pneumophila, include a…” and “Legionnaires’ Disease”
- Line 38: Reporting cases to the CDC is widely known to be an under-estimate of actual cases, rephrase sentence to demonstrate there were at least 10,000 cases or there were 10,000 cases reported but we know there are more.
- Line 96: “fecal pathogens to persist in”
- Lines 111-112: Referring to “Antibiotics…so cells can survive in biofilms.” The inability of antibiotics and disinfectants to penetrate a matrix is not the sole reason cells can survive in biofilms. It aids in the survival of cells, but there are many more reasons why cells flourish in biofilms. Consider revision or clarification.
- Lines 133-136: information after the ; is a sentence fragment. Please clarify
- Line 185: “would be of value to manufacturers”
- Line 197: “not enough disinfectant to kill”
- Line 243-244: “the broad resistance to antimicrobials”
- Line 318: Consider using 5-6% sodium hypochlorite instead of a brand name product
- Lines 319-320: “…to produce large drops that do not…”
- Line 325:
- “passage of OPPPs, as the pores are too large.”
- “In fact, as study at the EPA”
- Line 324: add “(GAC) filters” since the abbreviation is used later
Author Response
I thank this Reviewer for their detailed comments, excellent guidance for revision, and a reminder that patient care is the standard for driving monitoring, not simply cost.
Major Comnents.
- Yes, there is a bias in presentation towards the Mycobacterium spp, and that has been rectified by inclusion of citations for L. peumophila and P. aeruginosa in sections 2.2, 3.2, 3.3, 5.0, and 6.0). In part that bias was based on the fact that this chapter is part of the Journal's issue of "Legionella and Legionnaires' Disease: Pathogenesis, Prevention, and Public Health" and my assumption that I might duplicate information of L. pneumophila. That is clearly not the case of P. aeruginosa.
- The inappropriate and incorrect guidance on sampling biofilms only (Lines 125-128) has been deleted.
- The comments on whether growth of L. pneumophila can occur outside of amoeba (Lines 142-144) has been deleted.
- I have revised lines 151, 157-171, 181, 184, and 246 and in other sentences to delete first person language.
- Lines 176-179. I have revised the sentences to use "mutual exclusion" throughout, as that has been the conclusion of the reported research. Further, I have added the relevant information of the competitive relationship between Legionella and Pseudomonas in biofilms in lines 188-192 (e.g., new reference 45).
- Section 5 (Transmission) has been revised to include more details of the three modes of transmission and examples from OOOPs other than M. avium. These incliude aerosol transmission of P. aeruginosa and A. baumannii. The paragraph on water-borne transmission now includes citations and data on P. aeruginosa and S. maltophilia. The paragraph on transmission in homes has been expanded to include sponge-mediated transmission of A baumannii.
- Section 6 (Monitoring) has been revise to include patient care standards as a motivating factor, not just cost. Further, the fact that several jurisdictions (New York and New York City) and the Veterans Administration now require monitoring has been added.
Minor Comments
- Line 20 and following. The section numbers now are foll0wed by a period.
- Lines 60-86. The jump in numbering reflects the deletion of an individual section that was added to another, without changing the section numbers. the section numbers and subsection numbers have been corrected.
- L. pneumonia has been changed to L. pneumophilia throughout.
- Lines 10, 12, 33, 37, 38, and 96 have been corrected as suggested
- Lines 111-112. The discussion of the basis for antibiotic-resistance of biofilm cells has been increased and further documented.
- Lines 133-136. Sentence fragment has been corrected.
- Lines 185, 197, 243-244, 318, 319-320, and 324 have been corrected as suggested.
Round 2
Reviewer 2 Report
The author has addressed all of my provided comments and provided a thoroughly revised and acceptable manuscript.